# From Pixels to UI Actions: Learning to Follow Instructions via Graphical User Interfaces

**Peter Shaw**[1][*]     **Mandar Joshi**[1][*]     **James Cohan**[2]     **Jonathan Berant**[1]

**Panupong Pasupat**[1]     **Hexiang Hu**[1]     **Urvashi Khandelwal**[1]     **Kenton Lee**[1]

**Kristina Toutanova**[1]

[1] Google DeepMind     [2] Google

## Abstract

Much of the previous work towards digital agents for graphical user interfaces (GUIs) has relied on text-based representations (derived from HTML or other structured data sources), which are not always readily available. These input representations have been often coupled with custom, task-specific action spaces. This paper focuses on creating agents that interact with the digital world using the same conceptual interface that humans commonly use — via pixel-based screenshots and a generic action space corresponding to keyboard and mouse actions. Building upon recent progress in pixel-based pretraining, we show, for the first time, that it is possible for such agents to outperform human crowdworkers on the MiniWob++ benchmark of GUI-based instruction following tasks.

## 1   Introduction

Systems that can follow instructions to complete tasks through graphical user interfaces (GUIs) can help automate tedious tasks, improve accessibility, and expand the usefulness of digital assistants by allowing them to interact with tools and services. Despite the visual nature of GUIs, prior work has primarily focused on utilizing structured representations of the user interfaces (such as HTML sources, Document Object Model (DOM) trees, and Android view hierarchies) as well as custom, task-specific representations of high-level actions based on these structured representations (see §6). Recent efforts have achieved positive outcomes thanks to the advances of powerful language models (Gur et al., 2022; Kim et al., 2023; Yao et al., 2022).

While structured and task-specific representations may be useful, they are not always available – some examples are web applications that use extensive scripting, sandboxed environments where access to DOM is limited, and mobile applications which often do not expose the underlying structure to external modules. Even when structured application source data is available, it may be hard to interpret due to obfuscation and misalignment with what actually appears on the GUIs. Finally, aligning human demonstrations with task-dependent actions is often challenging.

In contrast, people interact with GUIs by perceiving the visual input and using generic mouse and keyboard actions, without needing to inspect the application's source code for cues on its functionality. They can quickly learn to interact with new applications that offer familiar visual interfaces, regardless of differences in implementation technologies. In this paper we ask: *Can we build an agent that can*

---

[*]Equal contribution.

37th Conference on Neural Information Processing Systems (NeurIPS 2023).

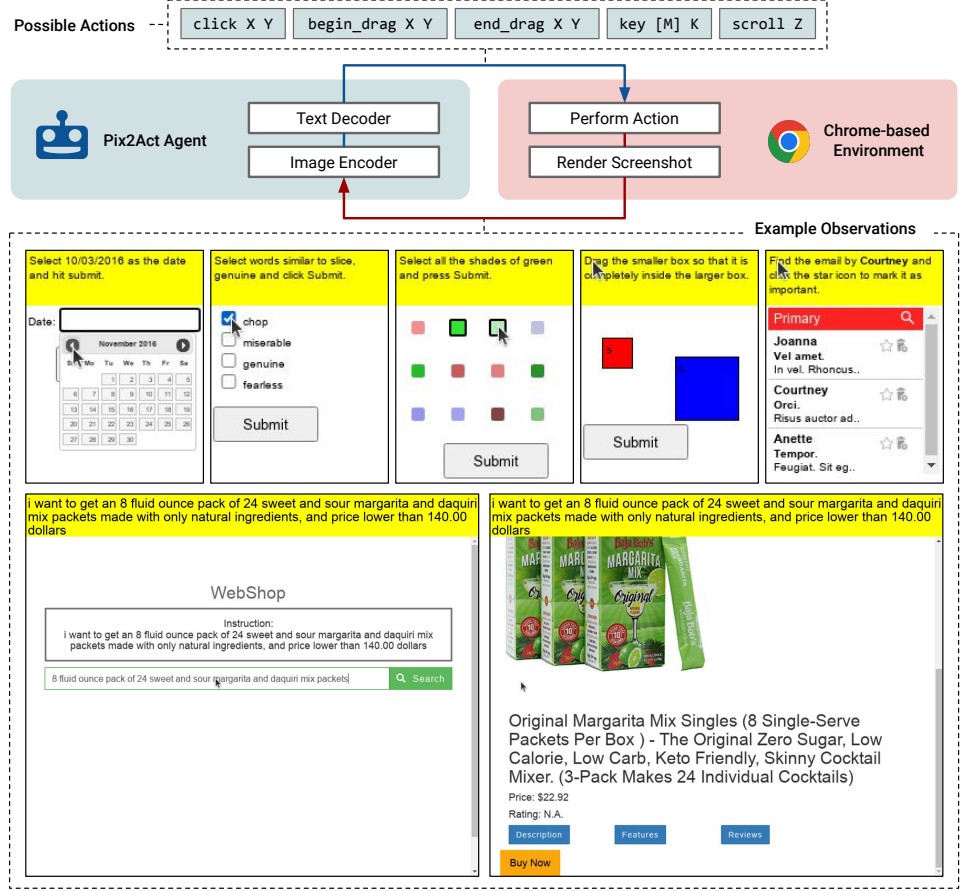

Figure 1: **Our agent learns to follow instructions via Graphical User Interfaces (GUIs).** Unlike most prior work studying instruction following for GUI-based tasks, our agent does not rely on text-based observations corresponding to DOM trees or HTML source code, or task-specific actions. Instead, our agent receives pixel-based observations and generates outputs corresponding to mouse and keyboard actions. The possible actions are encoded as text and shown on the top of the figure. We show examples of observations from various episodes for two benchmarks, MiniWob++ (top row) and WebShop (bottom row), that we adapt to study within the context of our general Chrome-based environment framework. See details in §2.

*complete tasks for users while relying solely on pixel-level visual representations of the GUI state, and generic low-level actions?*

Learning based on pixel-only inputs proved effective for game playing environments such as Atari (Mnih et al., 2015). However, for GUI-based instruction following tasks, learning from pixel-only inputs coupled with general low-level actions leads to several challenges. Interpreting GUIs visually requires understanding the interface layout, recognizing and interpreting visually-situated natural language, identifying visual elements, and predicting their functions and methods of interaction. A generic action space also poses the challenge of a more complex mapping between high-level textual instructions and corresponding sequences of low-level actions. As an example of the increased difficulty in this setting, on the MiniWob++ benchmark (Shi et al., 2017; Liu et al., 2018) of web GUI interaction, CC-Net (Humphreys et al., 2022) demonstrates human-level accuracy when accessing both screenshots and DOM structure, but its performance drops by 75% when the DOM information is removed from the agent's observations.

Here we present PIX2ACT, a model that relies solely on pixel-based screenshots as input and selects actions corresponding to basic mouse and keyboard functionalities.[2] We build on PIX2STRUCT (Lee et al., 2022), a Transformer-based (Vaswani et al., 2017) image-to-text model pre-trained to map

---

[2]Code and models are available at `https://github.com/google-deepmind/pix2act`.

screenshots to structured representations derived from HTML on web-scale data. PIX2ACT tunes this model using a combination of human demonstrations and environment interactions, applying tree search to iteratively generate new expert trajectories for training. We develop a general browser-based environment framework, and adapt two benchmark datasets, MiniWob++ and WebShop (Yao et al., 2022), to our setting with a unified, general purpose observation and action format.

On MiniWob++, PIX2ACT outperforms human crowdworkers and improves task score nearly 4x compared to the best prior results in our proposed setting (CC-Net without DOM). Ablations show that a key ingredient for PIX2ACT's performance is the pixel-based pre-training of PIX2STRUCT.

Our contributions are as follows:

1. We show, for the first time, that an agent using pixel-only inputs and a generic action space can outperform human crowdworkers on the MiniWob++ benchmark, significantly improving over prior work on this setting, and reaching performance comparable to that of state-of-the-art agents that access DOM information and use a comparable number of human demonstrations.

2. We adapt the WebShop benchmark to our setting, using pixel-based observations and general low-level actions. We establish the first baseline on this setting, although there is still a performance gap relative to larger language models using HTML-based inputs and task-specific actions.

3. We show that PIX2STRUCT's pre-training via screenshot parsing is effective for GUI-based instruction following with pixel-based inputs. In the behavioral cloning setting, pre-training improves task scores from 17.1 to 66.5 on MiniWob++ and from 1.1 to 46.7 on WebShop.

4. We demonstrate the successful application of tree search as a relatively simple method for policy improvement for MiniWob++.

## 2 Environment

Following the reinforcement learning literature, we model GUI interaction as a Markov Decision Process (MDP): at each time step, our agent receives an observation and selects an action. We develop a common environment framework with shared observation and action formats for browser-based tasks. Similarly to prior work on web-based agents (Liu et al., 2018), we use Selenium to programmatically interact with the Google Chrome browser.

**Observations** To form an observation, we first take a screenshot of the current browser window using Selenium and then augment it with additional information. First, if not already present, we render the natural language instruction on the top of the screenshot, following Lee et al. (2022). Second, as Selenium screenshots do not include cursors (which are typically rendered by the operating system), we draw a cursor on the screenshot to indicate the mouse pointer position. Finally, we render an indicator of whether the mouse button is currently pressed down, which is useful for dragging actions.

**Actions** Our action space consists of raw mouse and keyboard actions, as shown in Figure 1, where X and Y refer to discrete coordinate bins, K is one or more keys, M is an optional modifier key such as "shift", and Z refers to a vertical scroll amount, also represented as a discrete bin.[3] The `begin_drag` and `end_drag` actions can be used to execute "click and drag" actions. We use a configurable number of coordinate buckets per vertical and horizontal axis. Importantly, the DOM information is not provided by the environment and is therefore not used in any way to define observations or actions.

**Episodes and Rewards** Episodes continue until a terminal state or a configurable number of maximum steps is reached. For the environments we consider, the agent only receives a reward at a terminal state. This can be a binary reward based on whether the task was completed successfully or a partial reward based on how well the task was completed.

---

[3]We chose discrete bins because they enable a simple encoding of actions as tokens. Alternatives could include continuously-valued coordinates or relative movements with foveated binning (Baker et al., 2022).

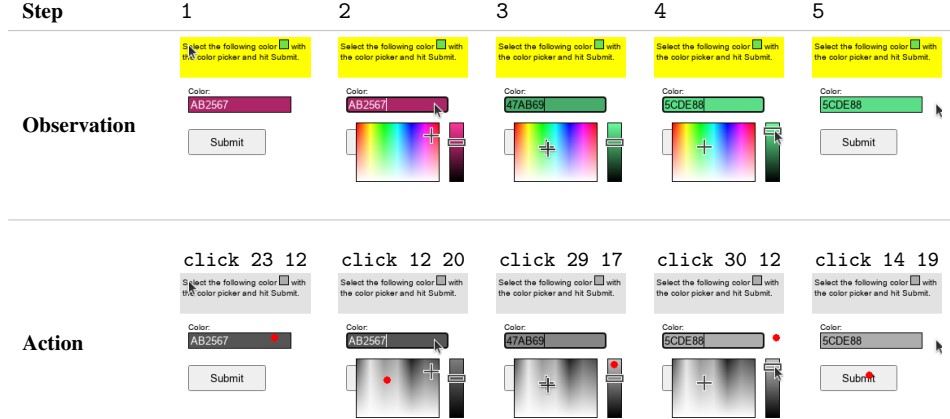

Figure 2: **An example episode of our agent on the MiniWob++ `use-colorwheel-2` task.** At each step, the agent receives a new observation and outputs the next action to take. The screenshots include a rendered instruction that the agent needs to follow to successfully complete the episode. For MiniWob++, we use 32 vertical and horizontal coordinate bins to specify locations. We show the click location visually for this figure.

## 3    Proposed Agent

Our agent, PIX2ACT, is based on the PIX2STRUCT model (Lee et al., 2022), which uses an image Transformer encoder and a text Transformer decoder. The architecture is based on Vision Transformer (Dosovitskiy et al., 2021) and T5 (Raffel et al., 2020). PIX2STRUCT is pre-trained on a *screenshot parsing* task: predicting simplified HTMLs from screenshots with visually-masked regions. Such pre-training was proven effective for tasks related to understanding user interfaces in a non-interactive setting, such as screen summarization and widget captioning (Wang et al., 2021; Li et al., 2020b). We use the PIX2STRUCT base variant with 282M parameters (12 encoder and 12 decoder layers; hidden size 768) for all our experiments. The model is called once per time step.

**Input**    The only input to the model is pixel-based observation from the environment. We can also condition on multiple previous observations by concatenating multiple frames. In preliminary experiments, we did not observe significant gains from conditioning on past observations for MiniWob++, and thus we only use the screenshot of the current step in our experiments. We reuse PIX2STRUCT's image processing by scaling input images up or down so as to extract the maximal number of fixed-size patches that still fit within the sequence length limit. We use resolutions of 160×210 and 800×600 for MiniWoB++ and WebShop, respectively.

**Output**    We encode actions as text tokens, which are predicted autoregressively by the Transformer decoder. We use beam search over tokens to output the $k$-best actions (see Appendix B.1 for details).

**Greedy Policy**    For interacting with the environment, we adopt a standard greedy policy, selecting the highest scoring action at each step, with one modification. To help prevent the agent from getting stuck in cycles, we track which actions have been taken for a given observation, and select the highest probability action in the beam that has not previously been taken given the current observation, which provides a modest increase in performance.

### 3.1    Training

We explore two methods for training models to follow instructions via GUIs. First, similarly to prior work, we use Behavioral Cloning (BC), where we train our model using standard supervised learning to predict the given action for each observation in a set of human demonstrations. Second, given access to environments with reward signals, prior work has also explored Reinforcement Learning (RL) to further improve agent performance. As an alternative to common reinforcement learning algorithms such as REINFORCE (Williams, 2004) and PPO (Schulman et al., 2017), we apply tree search as a simple method for policy improvement.

**Tree Search**   For a given set of model parameters, tree search leverages the deterministic nature of the environment to look ahead at the consequences of possible actions to determine a more optimal policy than greedily selecting actions.

We adopt Monte Carlo Tree Search (MCTS) (Coulom, 2006), which outperformed more naive search algorithms in initial experiments, and has been successfully integrated with neural network policies in prior work (Silver et al., 2017; Anthony et al., 2017). Similarly to this prior work, we train a model to estimate a *value function*, which predicts the value (i.e., estimated future rewards) of a given state. We use a surrogate reward which penalizes the number of steps taken to encourage concise trajectories without unnecessary actions. We implement this value function approximator using the same PIX2STRUCT architecture used for our agent.[4] However, instead of predicting actions, this model predicts state-values mapped to discrete buckets. To estimate the value of leaf states during MCTS, we use a combination of this value function approximator and rollouts using our greedy policy, similarly to Silver et al. (2017). See Appendix B for additional technical details.

We can then use successful episodes found with this stronger tree search policy to improve our model. As this stronger model then yields a more effective tree search policy, we can continue to iteratively improve our model using this method. Notably, this approach requires no modifications to the fine-tuning procedure of PIX2ACT, as, for simplicity, we tune on episodes from the tree search policy using standard supervised learning.

## 4   Benchmarks and Demonstrations

We adapt two benchmarks, MiniWob++ and WebShop, to our environment framework (§2) which consists of pixel-based observations and generic low-level actions. We also map previously collected human demonstrations for these benchmarks to our observation and action spaces.

### 4.1   MiniWob++

MiniWob++ (Liu et al., 2018) is a set of over a hundred web-browser based tasks. See Figures 1 and 2 for task examples. Each task consists of an algorithm for generating variations of the task and an instruction template, controlled by a random seed, with up to billions of possible configurations per task. The task instruction is given as (mostly) natural language text in the top yellow part, which in our framework can only be accessed visually. An automatic reward is given at the end of the task.

**Human Demonstrations**   We use the human demonstrations collected by Humphreys et al. (2022). However, their demonstrations were collected using an X11-based environment, which is different from our Selenium-based environment. This results in different renderings of the same underlying environment state, introducing a shift between the screenshots seen during training and those observed at test time. Additionally, we need to map from their real-time X11-based action sequences to our action space. We were able to perform this mapping with a reasonable degree of success for 59 tasks. Notably, not all behaviors in the human demonstrations are supported in our Selenium-based environment. For example, Selenium does not implement the ability to highlight text and drag it into a text field, and such an action is widely used in the human demonstrations for tasks where text is copied and pasted. Additionally, while our environment framework intends to cover the basic functionality of most web interfaces, aspects of some MiniWob++ tasks, such as capturing real-time observations for animated elements, are not supported. See Appendix A for additional details.[5]

Starting with approximately 1.3 million demonstrations across the 59 supported tasks, we filtered demonstrations with a reward of $< 0.8$, or approximately 6% of demonstrations. We were able to successfully convert 81% of the remaining demonstrations to our action space. We reserve 10% of the data for a development set. Demonstrations contain approximately 3 steps per task on average, although this varies considerably across tasks.

---

[4]While it may be more efficient to share an encoder between these two PIX2STRUCT-based models that condition on the same inputs, we trained separate models for simplicity.

[5]Other prior work has used the demonstrations from Liu et al. (2018), which cover a different subset of MiniWob++ tasks. However, these demonstrations do not include screenshots or sufficient information to replay the episodes in a browser environment to collect new screenshots, and therefore cannot be applied to our setting.

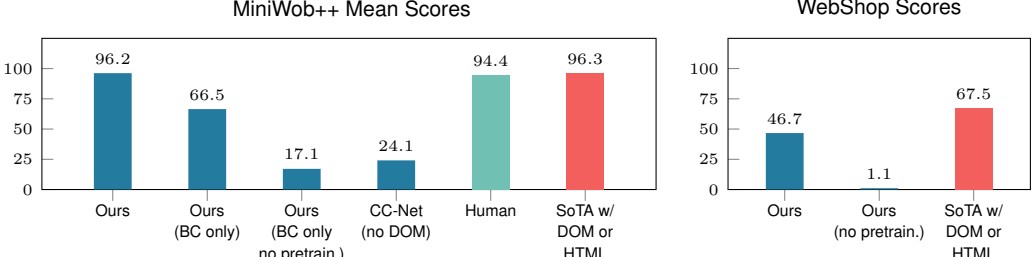

Figure 3: **Main results evaluating PIX2ACT (ours) on MiniWob++ (left) and WebShop (right)**. In this paper we focus on approaches that do not have access to DOM or HTML information, and recieve pixel-based observations (**blue**). On this setting, PIX2ACT significantly improves over prior work on MiniWob++ and establishes the first baseline on WebShop. Our method performs competitively with humans (**green**) and with methods that have access to DOM or HTML information (**red**) on MiniWob++, although there is a gap with the best performing methods that access HTML on WebShop (see §5.3 for detailed analysis).

**Evaluation** We report the mean score across seeds and tasks. The score is the MiniWob++ raw reward (without time decay) mapped from the original range $[-1, 1]$ to the range $[0, 100]$. The score is equivalent to the success rate (*i.e.* the proportion of episodes in which the agent receives a positive reward) for tasks with binary rewards. For episodes that do not complete due to reaching a maximum number of allowed steps, we assume a score of 0. For each task, we compute the mean over 100 random seeds, and then compute the mean over 59 MiniWob++ tasks.

### 4.2 WebShop

WebShop (Yao et al., 2022) is a web-based shopping environment with over 1.1 million products from Amazon. The task is to find and purchase a product based on a human-authored text instruction. Finding a suitable product requires entering search queries, clicking on results, and determining the relevance of various products to the instruction. An automatic reward is computed based on similarity between the purchased product and the gold target product.

**Human Demonstrations** We use the 1,566 human demonstrations (with a train/development/test split of 1012/54/500) collected in Yao et al. (2022). As with the MiniWob++ demonstrations, we need to map between the observation and action sequences used in their setup to our framework. Yao et al. (2022) used high-level actions (*e.g.* "search" or "click[item]"), each of which could map to multiple lower-level actions in our environment. Specifically, for all actions involving a mouse click, we determine the coordinates of the center of the corresponding HTML element. For WebShop, the entire screen content is not always visible due to page heights exceeding the viewport dimensions. If the clicked element lies outside the visible area, we add scroll actions until the element is visible. Finally, we map search actions to two actions in our environment: clicking on the center of the search box and entering the search query followed by the *enter* key. We render the HTML inputs in the human demonstrations using our browser to obtain screenshots. Additionally we found that rendering the last 5 actions (separated by ) on top of the screenshot to be helpful.

**Evaluation** Consistent with previous work, we report Task Score, which is the average reward across 500 test instructions.

## 5 Experiments and Analysis

### 5.1 Training Details

We updated all model parameters during fine-tuning, including both the image encoder and text decoder. We used the Adafactor optimizer (Shazeer and Stern, 2018) with a learning rate of 0.01.

**MiniWoB++** We finetuned a single model jointly on episodes from all tasks for a total of 26K steps using a batch size of 512, input/output sequence lengths of 512/16. We also evaluated using the tree search procedure described in §3.1 to improve our agent. We performed 2 iterations of policy

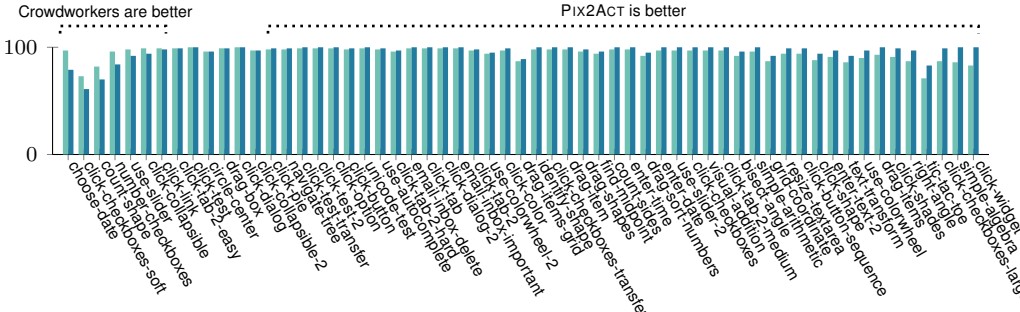

Figure 4: Comparing scores on MiniWob++ tasks of PIX2ACT (**blue**) with human crowdworkers (**green**), ranked from left to right by the relative difference in performance.

improvement with tree search, collecting a total of 826K episodes across all tasks, and tuning for a further 26K steps.

**WebShop** We used only the provided human demonstrations to train our model.[6] Due to its larger resolution and text-heavy data, we used a higher input sequence length of 4096. We also found it useful to perform intermediate finetuning on MiniWoB++, followed by 10K steps of further finetuning on WebShop using a batch size of 256 (see §5.3 for details).

## 5.2   Main Results

We report the results of our models on MiniWob++ and WebShop in Figure 3. For MiniWob++, we also provide task-level comparisons between PIX2ACT and human crowdworkers in Figure 4. There is limited prior work studying these tasks without access to DOM and HTML information. For MiniWob++, the only comparable baselines are from the CC-Net model of Humphreys et al. (2022), which mentions an ablation experiment where performance dropped by 75% from their primary results when the models conditioned on only screenshots without DOM information. As they did not provide per-task numbers for this ablation, we estimate the performance of CC-Net without DOM information by assuming that the drop in performance on the subset of tasks we study was also 75%. Regardless, it is clear that PIX2ACT significantly outperforms CC-Net on this setting. The difference in performance can be largely attributed to the screenshot parsing pre-training of Lee et al. (2022). For WebShop, there is no prior work exploring such a setting, so we establish the first baseline.

## 5.3   Ablations and Analysis

**Pre-training ablations** To study the impact of the pre-training on our model's ability to effectively learn to follow instructions via GUIs, we evaluate model performance without the pre-training procedure. For these experiments, we only compared performance of models trained using behavioral cloning. The results are shown in Figure 3, and demonstrate that pre-training is critical for our model's performance.

**Comparison with models that use DOM or HTML as input** We can also compare our results without access to DOM or HTML to previous methods that utilized these resources, including those which also leverage DOM information to construct specialized action spaces. The performance of the best model from prior work leveraging DOM or HTML information is shown in Figure 3.

For MiniWob++, the best model on this setting is CC-Net (Humphreys et al., 2022) trained with BC and RL and with access to both DOM and pixel-based observations.[7] PIX2ACT achieves comparable performance to their best model, while relying on only a subset of the information used by CC-Net, and using a comparable number of human demonstrations for training. PIX2ACT also outperforms

---

[6]We did not explore applying RL techniques to WebShop in this work. Prior work (Yao et al., 2022) has not shown as significant an advantage to applying RL on WebShop relative to the large improvements shown by prior work on MiniWob++, which offers a near limitless variety of environments with reward signals for training.

[7]We compute mean scores for CC-Net by averaging their reported per-task results over the 59 tasks we study.

| Pre-training | Included | Heldout |
|---|---|---|
| Yes | 65.5 | 28.3 |
| No | 11.0 | 7.6 |

Table 1: We selected 9 MiniWob++ tasks and evaluated mean scores when they are *heldout* from the training set. Pretraining leads to non-trivial generalization (28.3) to held out tasks that were unobserved at training time compared to a randomly initialized model (7.6). We also include scores when the tasks are *included* during training for reference.

| | | Iteration | |
|---|---|---|---|
| Policy | 0 | 1 | 2 |
| Greedy | 66.5 | 93.1 | 96.2 |
| Tree Search | 91.7 | 98.4 | — |

Table 2: We compare average MiniWob++ scores using the greedy policy with one that uses tree search and lookahead, given the same underlying model. The model is initially trained on human demonstrations and iteratively improved by training on episodes generated by the tree search policy.

CC-Net when each model is trained only with behavioral cloning, as CC-Net performance on this setting drops to 38.7 (results not shown in the Figure). Notably, CC-Net scores also drop by approximately 10% when the model is not given access to a dictionary of input strings provided by the environment. As shown in Figure 3, the key to our model's ability to achieve comparable performance without relying on DOM-based inputs is pixel-based pre-training. Another difference is that CC-Net uses a real time setting, which enables some forms of interaction not supported by our environment, and therefore can support a larger set of MiniWob++ tasks. On the other hand, for BC, CC-Net does not need to handle the shift in rendering format and potentially noisy action space conversion.

For WebShop, the best model on this setting is WebGUM (Furuta et al., 2023a), which leverages the HTML source, a custom action space for the shopping domain, and a Flan-T5-XL (Chung et al., 2022) backbone. WebGUM outperforms PIX2ACT when compared on this setting. Some of this gap can be attributed to their simplified high-level action space, direct access to the relevant text on the page, and ability to transfer from Flan-T5's pretraining scale and instruction finetuning. Comparable improvements to the scale and pretraining of pixel-based models could reduce this gap.

We discuss other approaches that leverage DOM or HTML information further in §6. We also offer a complete comparison across all MiniWob++ tasks in Appendix C.

**Evaluating transfer across tasks**    Training a pretrained, pixel-based model to interact with a GUI can intuitively lead to better generalization to new tasks that use common GUI design principles. To study this, we evaluate the ability of PIX2ACT (without RL) to generalize to tasks unseen during training. Specifically, we hold out 9 out of 59 tasks and train on the remaining 50.[8] We then evaluate performance on the held-out tasks, comparing initializing with PIX2STRUCT to random initialization. Table 1 illustrates that PIX2ACT can reach a mean score of 28.3 on held out tasks compared to 65.5 when training on those tasks. Conversely, mean score is 7.6 when PIX2STRUCT initialization is not used. This shows that combining pretraining with a general GUI interface can lead to non-trivial generalization to held out tasks.

For WebShop, we find that finetuning directly on WebShop (without intermediate finetuning on MiniWoB++ as mentioned in 5.1) results in a drop of 4.0 in Task Score, demonstrating transfer learning benefits across these datasets.

**Tree search analysis**    Table 2 shows the improvement in MiniWob++ scores by training on episodes generated by tree search. After an initial round of training on episodes generated by tree search, the effectiveness of tree search also improves due to improvements in the underlying model used to guide the search. The best greedy policy achieves performance close to the best tree search policy, but does not require access to reward signals or additional exploration at inference time. Our results indicate that we could further improve performance with more iterations of policy improvement via tree search.

---

[8]We manually pick the 9 tasks to verify they include only actions or elements that would be reasonable to generalize to from the training tasks. The tasks are `click-checkboxes-large`, `click-color`, `click-tab-2`, `click-tab-2-hard`, `count-shape`, `drag-shapes`, `use-color-wheel-2`, `use-slider-2`.

## 6 Related Work

We focus on agents that interact with GUIs, such as operating system dialogs or web pages, to accomplish a given task. Many early approaches relied on the structured information from the GUIs (Zettlemoyer and St. Amant, 1999; Allen et al., 2007; Branavan et al., 2010). This information could range from a flat list of GUI components and their properties, to the full hierarchical structure of the components (*e.g.* the DOM tree). The output space also depends on this structured information, often using GUI components as action targets (*e.g.* clicking button #7). As discussed in §1, such structured information might not always be available, or might not align with what visually appears to the users.

When Shi et al. (2017) introduced the *World of Bits* tasks, which was the precursor to MiniWob++ (Liu et al., 2018), they proposed a model based on a convolutional neural network that takes both visual and structured inputs and then performs generic low-level computer actions (*e.g.* clicking at a coordinate or pressing a key), similarly to PIX2ACT. However, the model performed poorly compared to humans. Follow-up work studied specialized architectures for incorporating structured DOM information and restricted the action space to clicking and typing predetermined texts on DOM elements (Liu et al., 2018; Gur et al., 2018; Jia et al., 2019). Humphreys et al. (2022) reconsidered incorporating both visual and structured information as well as a low-level action space that aligns better to the human demonstrations. We discussed their approach, CC-Net, in §5.3. Humphreys et al. (2022) also explored the benefits of large-scale human demonstrations, and we build on their work to utilize a large number of human demonstrations to train PIX2ACT. This paper shows that PIX2ACT, a model with pixel-only inputs, can outperform humans on MiniWob++ and match the state-of-the-art approaches that rely on DOM information.

Automating web-based tasks using large language models (LLMs) has also been broadly explored. For instance, WebGPT uses a text-based web browsing environment to search and navigate the web (Nakano et al., 2021). More relatedly, recent work has investigated prompting LLMs to produce agents that can generalize to tasks based on a small number of in-context examples. Yao et al. (2023) proposed ReAct, a few-shot prompted LLM, which uses observations derived from HTML and a custom action space to make predictions based on explicit reasoning steps. Similarly, Kim et al. (2023) proposed RCI, a prompted LLM that iteratively critiques and refines its outputs, also using HTML inputs and custom action spaces. These approaches achieve competitive performance on WebShop and MiniWob++, respectively, and are extremely sample-efficient, relying on just a handful of demonstrations per task. Gur et al. (2022) treated raw HTML as a string and fed it to LLMs pretrained on natural language. After fine-tuning them on demonstrations, the models improved MiniWob++ task success rate and sample efficiency compared to models that take DOM-based inputs and specialized architectures. Finally, WebGUM (Furuta et al., 2023b), discussed in §5.3, extends HTML-based models to integrate a vision encoder pretrained on ImageNet-21K.

Other work has focused on tasks related to mobile apps. Li and Li (2022) considered a model with pixel-based inputs similar to that of Lee et al. (2022), and included evaluations on tasks related to grounding instructions to screenshots, but did not consider interactive environments. Some work has considered instruction following tasks in mobile app environments (Li et al., 2020a; Burns et al., 2022), but has generally not studied observation and action formats similar to ours, instead relying on inputs based on the Android view hierarchy. We focused on web-based GUIs so that we could use a consistent environment framework for simplicity. Besides GUIs, several works on video game agents also considered visual-only input and low-level actions. For example, most works on Atari games used the screenshot as visual input and predicted the controller buttons to press (Mnih et al., 2015). More recently, Baker et al. (2022), which focuses on learning from unlabeled videos, proposes an agent for Minecraft that uses pixel-based inputs paired with keyboard and mouse actions, similarly to PIX2ACT.

## 7 Limitations and Discussion

**Pixel-based vs. text-based representations**   Text-based representations may be practically useful when available, especially since they enable transferring knowledge from LLMs, demonstrating impressive few-shot learning with LLMs for MiniWob++ (Kim et al., 2023) and WebShop (Yao et al., 2023). When structured source is not available, OCR systems and models trained to predict the location and function of UI elements may also help connect models with the power of LLMs. On the other hand, similar advances in scaling and pre-training of vision or multimodal models could

potentially enable similar capabilities in a pixel-based setting in the future, as we have shown the effectiveness of pixel-based pre-training (albeit at a smaller scale) for GUI-based tasks. Nevertheless, beyond addressing the case where HTML or DOM information is unavailable, we hope our study contributes towards a better understanding of the potential of pixel-based representations for instruction following via GUIs.

**Tree Search**    Our approach to policy improvement with tree search for MiniWob++ relied on the ability to procedurally generate new MiniWob++ environment and instruction variations and receive reward signals for task completion. Both aspects are unlikely to be available for some real world environments, and such an approach might need to rely on generative models of potential instructions and approximate reward models for task completion (*e.g.* Bahdanau et al. (2018); Du et al. (2023)). Our implementation also relied on the ability to reset the environment to an initial state, a useful feature for environments being used for exploration. Additionally, while we show that tree search can be sufficient to reach high performance on MiniWob++, we did not perform a detailed comparison relative to other search and RL algorithms in this study, which would be useful to better understand the most efficient approaches for learning from GUI-based environments.

**Broader Impact**    In this paper we have trained and evaluated models only in offline environments. Responsibly deploying models in an environment where they can interact with online services would require additional considerations. Prior to enabling a model to access a new service, it would be important to sufficiently verify and/or constrain the behavior of the model to ensure that it is consistent with the terms-of-service for that service and does not otherwise cause harm. Ensuring sufficient data privacy could also be an important consideration for deploying models such as PIX2ACT that rely on capturing screenshots from browsers.

There would be many potential risks associated with deploying models that could interact with services in violation of their terms-of-service or otherwise engage in various forms of spam, fraud, or abuse. Examples of such behavior could include impersonating human users, generating harmful content or spam, or engaging in denial-of-service attacks. Models that use the same conceptual interface humans use could potentially be more capable of breaking security defenses (e.g. solving CAPTCHAs) or engaging in forms of spam, fraud, or abuse that are more difficult to detect. It is therefore important for research related to security and techniques for detecting spam, fraud, and abuse to take such potential uses into account.

## Acknowledgments

We would like to thank Peter Humphreys, Toby Pohlen, and Gregory Thornton for their assistance with the MiniWob++ demonstraions. We also thank Ming-Wei Chang, Austin Huang, Luheng He, Tianze Shi, David Gaddy, Jacob Eisenstein, and Yi Luan for useful discussions and comments.

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

# A Additional Dataset Details

## A.1 MiniWob++ Supported Tasks

MiniWob++ consists of 104 tasks. Most prior work (Shi et al., 2017; Liu et al., 2018; Gur et al., 2018; Jia et al., 2019) has evaluated performance on only a subset of these tasks, with the notable exception of Humphreys et al. (2022), which evaluated on all 104 tasks. We evaluated on 59 of these 104 tasks, based on our best effort attempt to (1) design a general purpose set of actions that could be implemented using Selenium and (2) convert the demonstrations collected by Humphreys et al. (2022) to our observation and action format. While further development of the conversion process and Selenium-based actions could potentially support more tasks, the 59 tasks we support still include a wide range of instructions and interactions. Note that determining the set of 59 tasks was based solely on the feasibility of conversion to our observation and action format, and *not* based on model performance. Below we offer further details.

Several tasks in MiniWob++ feature animated elements. These tasks can require sampling observations in a real-time manner in order to capture the information needed to select the correct action. Also, the effects of an action may be delayed and therefore not captured by an observation sampled immediately after the action has executed. MiniWob++ provides a `-nodelay` version for several tasks which removes such animations. We train and evaluate on the `-nodelay` version of these tasks (`choose-date`, `click-collapsible-2`, `click-collapsible`, `click-pie`, `use-autocomplete`). We exclude `choose-date-easy` and `choose-date-medium` which offer simpler versions of `choose-date` but do not have a corresponding `-nodelay` version. Additionally, we exclude `chase-circle`, `drag-cube`, `moving-items`, and `simon-says`, which feature animation without a `-nodelay` version.

Many MiniWob++ tasks also involve vertical scrolling. In the human demonstrations, this can be implemented using a scroll wheel, or various clicking or dragging interactions with a vertical scroll bar rendered on the right side of a scrollable element. Mapping such interactions to actions that lead to equivalent scrolling in our Selenium-based environment is non-trivial. Therefore, for simplicity, we excluded tasks that involve scrolling: `book-flight`, `click-scroll-list`, `email-inbox`, `email-inbox-nl-turk`, `read-table`, `read-table-2`, `scroll-text`, `scroll-text-2`, `search-engine`, `social-media`, `social-media-all`, `social-media-some`, `terminal`.

Demonstrations for many MiniWob++ tasks also include copying and pasting text. In many cases, this was executed in the human demonstrations by double clicking a text string and then clicking and dragging it into an input field. Such an interaction is not supported in Selenium, which made it challenging to support these tasks. This led us to exclude the following tasks: `login-user-popup`, `copy-paste`, `copy-paste-2`, `email-inbox-forward`, `email-inbox-forward-nl`, `email-inbox-forward-nl-turk`, `email-inbox-noscroll`, `email-inbox-reply`, `email-inbox-star-reply`, `enter-password`, `enter-text`, `enter-text-dynamic`, `find-word`, `login-user`, `multi-layouts`, `multi-orderings`.

Finally, we excluded several other tasks for various other reasons. The `choose-list` task uses the HTML `<select>` tag to implement a drop-down menu, which is not supported properly by our Selenium-based environment. The `click-menu` and `click-menu-2` tasks require unsupported mouseover effects. Demonstrations for the `text-editor` task features click and drag interactions to highlight text which do not have the same effect when executed in Selenium. There also appeared to be differences in how Selenium implemented the number input field for `guess-number`. Finally, we excluded several tasks due to low demonstration conversion success rates (`focus-text`, `focus-text-2`, `use-spinner`). Upon further investigation, this was due to episodes completing immediately after a "pointer down" event without a complete click for `focus-text` and `focus-text-2`, and due to frequent double clicking for `use-spinner`.

## A.2 MiniWob++ Rendering Differences

There are differences between the rendering of observations in the human demonstrations from Humphreys et al. (2022) and the rendering of environment state in our Selenium-based environment. We show an example in Figure 5, which shows subtle differences, *e.g.* in font style and in element sizes and positions.

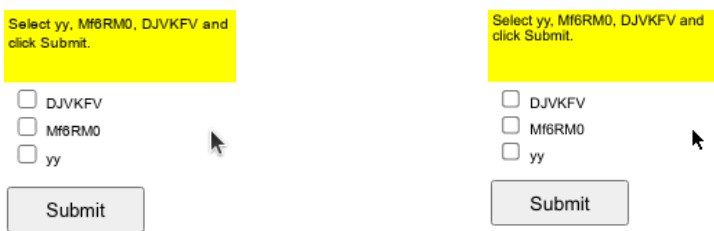

Figure 5: Comparison of differences between the screenshots of the human demonstrations for MiniWob++ from Humphreys et al. (2022) (right) with how the same environment state is rendered in our Selenium-based environment (left).

# B  Additional Technical Details

## B.1  Beam Search

As mentioned in §3, we use beam search over tokens in the text decoder to produce a set of top-$k$ actions for a given state, along with their approximate probabilities. We refer to these as approximate probabilities because they are subject to a length normalization factor (Wu et al., 2016) of 0.6 during beam search, following Raffel et al. (2020). For MiniWob and WebShop, our experiments used $k = 8$ and $k = 10$, respectively.

## B.2  Tree Search

Here we describe the details of the tree search approach described in §3.1. We adopt Monte Carlo Tree Search (MCTS) (Coulom, 2006), and follow prior work which has integrated MCTS with neural networks (Silver et al., 2017; Anthony et al., 2017), which we apply to MiniWob++ environments. We performed a minimal amount of tuning to determine an approach that yielded improvements in mean score over the greedy policy, even for the most challenging tasks.

**Problem Setting**  We consider an environment with states $\mathcal{S}$ and actions $\mathcal{A}$. The reward function, $r(s)$, returns a scalar corresponding to the reward given for transitioning to state $s \in \mathcal{S}$, and is described below. MiniWob++ environments are randomly generated, but transitions are deterministic within an environment generated by a particular random seed. The transition function, $f(s, a)$, returns the state resulting from taking action $a \in \mathcal{A}$ in state $s \in \mathcal{S}$.

**Surrogate reward**  Rather than using the raw reward directly provided by the MiniWob++ environment, we consider a surrogate reward: $r(s) = \alpha_s + r^t(s)$, where $\alpha_s$ provides a small negative reward that encourages shorter trajectories without unnecessary actions. $r^t(s)$ is the raw reward from the MiniWob++ environment if $s$ is a terminal state and the raw reward is $> 0.8$, or 0 otherwise. We use $\alpha_S = -\frac{1}{30}$. As all tasks can be completed within 30 steps, this is small enough to ensure a positive reward is possible for all tasks. Additionally, the penalty is small enough such that in practice the agent should not be incentivized to sacrifice raw reward to reduce the number of steps taken.

**Value network**  The value function $v^\pi(s)$ for a given policy $\pi$ is the expected future rewards from state $s$ if actions are selected according to policy $\pi$. The optimal value function, $v^*(s)$, is the expected future rewards if optimal actions are chosen. We attempt to learn an approximation of this function, $\hat{v}_\phi(s) \approx v^*(s)$, parameterized as a PIX2STRUCT-initialized model with parameters $\phi$, which we refer to as the *value network*. The model is trained on transitions from the human demonstrations, which demonstrate close to optimal behavior in many cases. For every state in the human demonstrations, we compute the actual future rewards for the given episode, according to the surrogate reward. We map these future rewards to discrete bins and represent them as integers in the PIX2STRUCT decoder. At inference time, we approximate the mean of the distribution over these discrete bins by considering the top-$n$ predictions from the model using beam search (with $n = 3$), weighted proportional to their respective probabilities.

**Policy network**  For consistency with prior work, we will refer to the PIX2STRUCT model tuned to generate actions (*i.e.* PIX2ACT) as the *policy network*, with parameters $\theta$. The greedy policy $\pi_\theta(s)$ selects the action $a$ with the highest approximate probability $p_\theta(a|s)$ in the top-$k$ beam (see §B.1), subject to the conditions described in §3.

**Search policy**  We can use lookahead search to implement a policy, $\pi_\theta^*(s)$, which leverages interactions with the environment ($f(s, a)$ and $r(s)$) to select actions in a more optimal way than the greedy policy $\pi_\theta(s)$. Both the policy network and value network are used to constrain and prioritize the search.

MCTS performs $K$ rounds of traversing a search tree with nodes corresponding to states, and edges corresponding to actions. Due to the computational cost of the policy and value networks, we use a modest number of rounds, $K = 16$, for our experiments. The search tree is initialized with a single root node for state $s$. Each round starts at $s$ and traverses the tree. At each step $t$ of a given round, an action $a_t$ is selected for state $s_t$, where $a_t = \max_a Q(s_t, a) + U(s_t, a)$. $Q(s_t, a)$ is an average reward over all rounds that have traversed the associated edge. It is based on actual accumulated rewards during tree traversal and the value estimates of leaf states (described below). $U(s_t, a) = c * p_\theta(a|s) * \frac{\sqrt{N(s_t)}}{1 + n(s_t, a)}$ is a term that encourages exploration, where $n(s_t, a)$ is the number of times action $a$ has been selected from state $s_t$, $N(s_t)$ is the total number of times state $s_t$ has been visited, and $c$ is a scalar hyperparameter that we set to $0.1$. Following Silver et al. (2017), we use the policy network to bias this exploration term. To constrain the search, we only consider the top-$k$ actions according to the policy network, where $k = 8$ in our experiments.

If we select an action $a_t$ for state $s_t$ which has never been previously selected from $s_t$, then the simulation ends and we add a new leaf state, $s_L = f(s_t, a)$, to the search tree. If $s_L$ is not a terminal state, then we estimate its value (*i.e.* future returns) using both the value network and a rollout with the greedy policy. Specifically, following Silver et al. (2017), we estimate its value as $\lambda * \hat{v}_\phi(s_L) + (1 - \lambda) * v^{\pi_\theta}(s_L)$ where $v^{\pi_\theta}(s_L)$ is equal to the actual returns from following the policy $\pi_\theta$ starting at $s_L$ for a maximum of 20 steps, with actual returns clipped to a minimum value of 0. Is there $\lambda$ is a mixing parameter that we set to $0.1$. For challenging environments, rollouts may be unlikely to find a terminal state with positive reward, and in such cases rollouts may not be very informative. On the other hand, the value network can provide poor value estimates for certain states, especially if they are not well represented in the human demonstrations. By combining both methods we aim to provide a better approximation of the value of leaf states. Returns are propagated up the tree to each parent $s'$ to update $Q(s', a)$. As $Q(s_L, a)$ is undefined prior to selecting $a$ from $s_L$ for the first time, we initialize $Q(s_L, a)$ for each action to be equal to the initial value estimate of $s_L$ plus $\alpha_s$.

To understand the impact of rollouts and value estimates using the value network, in Table 3 we compare mean scores over 12 challenging MiniWob++ tasks for different values of $\lambda$: 0 (rollout only), 0.1 (both rollout and value network), and 1 (value network only). We also include the mean score using the greedy policy for reference. These results use the policy network and value network trained on the human demonstrations. The results show that using a combination of rollouts and the value network gives the best results. The value network is primarily useful for challenging tasks that require longer trajectories, such as `number-checkboxes`, relative to using rollouts only.

| Greedy Policy | $\lambda = 0$ (rollout only) | $\lambda = 0.1$ | $\lambda = 1$ (value network only) |
|---|---|---|---|
| 28.8 | 74.2 | **78.3** | 57.4 |

Table 3: Mean scores for different policies over 12 challenging MiniWob++ tasks.

Once we have completed $K$ rounds, $\pi_\theta^*(s)$ selects the most visited action $a$ for state $s$, and we begin the process again at the subsequent state. We reuse the search tree for subsequent time steps for efficiency, so we require only $K - n(s, a)$ additional rounds for the subsequent state.

**Policy improvement**  We can sample trajectories with $\pi_\theta^*$, then update $\theta$ by training $\pi_\theta(s)$ to approximate $\pi_\theta^*(s)$ for each $s$ in the sampled trajectories. This then also improves $\pi_\theta^*(s)$, as $\theta$ informs how the search space is constrained and prioritized. Therefore, we can continue to iteratively improve $\pi_\theta(s)$. To produce these trajectories, we randomly sample MiniWob++ tasks and seeds, and select

actions according to $\pi_\theta^*$. We then filter trajectories where the raw reward is $< 0.8$. We then tune $\theta$ on these new trajectories. For simplicity, we keep the value network (*i.e.* $\phi$) fixed.

We initially found that tuning on trajectories from MCTS could be unstable, leading to an early loss spike. To resolve this, we slightly decreased the learning rate (from $1e - 3$ to $5e - 4$) and increased the number of warmup steps (from 1000 to 4000) relative to the hyperparameters used for behavioral cloning.

### B.3 Compute Details

We fine-tuned models using 64 Google Cloud TPU v3 cores.

## C Additional Results

### C.1 Variance Estimates

We evaluated results for MiniWob++ based on 100 randomly selected seeds for each of the 59 tasks. To understand how results vary depending on which 100 seeds per task are used for evaluation, we ran 3 trials with different evaluation seeds for the strongest PIX2ACT model reported in Table 3, yielding mean scores of 96.2, 96.4, and 96.1; the standard deviation across these trials was 0.15. For WebShop, there is a standard test set consisting of 500 instances, so selecting seeds for evaluation is not necessary.

### C.2 MiniWob++ Results Per Task

We show the performance of PIX2ACT (ours) on each of the 59 MiniWob++ tasks we study, compared to other approaches, in Table 4. We compare with human crowdworker performance reported by Humphreys et al. (2022), CC-Net (Humphreys et al., 2022), DOM-Q-Net (Jia et al., 2019), DOMNET with workflow-guided execution (Liu et al., 2018), QWeb (Gur et al., 2018), RCI (Kim et al., 2023), WebN-T5-3B (Gur et al., 2022), and WebGUM (Furuta et al., 2023a). We also report scores for PIX2ACT and CC-Net with behavioral cloning (BC) only. We do not include scores for GlobalCNN (Shi et al., 2017), which reported only human normalized success rates. Other than Humphreys et al. (2022), prior work has primarily reported success rate (*i.e.* the percentage of episodes with positive rewards), which can be equivalently mapped to the scores we report for tasks without partial rewards.

| Task | Ours | Ours (BC) | Human | CC-Net | CC-Net (BC) | DOMNET | DOM-Q-Net | QWeb | RCI | WebN-T5 | WebGUM |
|------|------|-----------|-------|--------|-------------|--------|-----------|------|-----|---------|--------|
| bisect-angle | 96 | 32 | 92 | 97 | 29 | — | — | — | — | — | — |
| choose-date | 79 | 6 | 97 | 97 | 12 | 0 | 100 | — | — | 0 | 13 |
| circle-center | 96 | 52 | 96 | 97 | 36 | — | — | — | — | — | — |
| click-button | 99 | 32 | 98 | 100 | 78 | 100 | 100 | 100 | 100 | 100 | 100 |
| click-button-sequence | 99 | 100 | 94 | 100 | 47 | 100 | 100 | — | 100 | 100 | 100 |
| click-checkboxes | 100 | 99 | 97 | 98 | 32 | 100 | 100 | — | 100 | 96 | 100 |
| click-checkboxes-large | 99 | 100 | 87 | 71 | 0 | 84 | — | — | 94 | 22 | 99 |
| click-checkboxes-soft | 61 | 91 | 73 | 95 | 4 | 94 | — | — | 72 | 54 | 98 |
| click-checkboxes-transfer | 100 | 76 | 98 | 99 | 36 | 64 | — | — | 100 | 63 | 99 |
| click-collapsible-2 | 97 | 31 | 97 | 98 | 17 | 99 | — | — | 62 | 0 | 95 |
| click-collapsible | 94 | 80 | 99 | 100 | 81 | 100 | — | 100 | 100 | 0 | 98 |
| click-color | 99 | 88 | 97 | 100 | 82 | 100 | — | — | 100 | 27 | 34 |
| click-dialog | 100 | 12 | 100 | 100 | 95 | 100 | 100 | 100 | 100 | 100 | 100 |
| click-dialog-2 | 100 | 73 | 99 | 100 | 88 | 100 | — | — | 100 | 24 | 43 |
| click-link | 98 | 86 | 99 | 99 | 59 | 100 | 100 | 100 | 100 | 100 | 100 |
| click-option | 100 | 0 | 99 | 99 | 21 | 100 | 100 | — | 100 | 87 | 100 |
| click-pie | 99 | 81 | 98 | 97 | 15 | 32 | — | 100 | — | 51 | 99 |
| click-shades | 99 | 76 | 91 | 100 | 4 | 99 | — | — | 100 | 0 | 0 |
| click-shape | 94 | 19 | 88 | 95 | 11 | 64 | — | — | 98 | 53 | 72 |
| click-tab | 100 | 54 | 99 | 100 | 95 | 100 | 100 | 100 | 100 | 74 | 100 |
| click-tab-2 | 98 | 42 | 97 | 98 | 27 | 98 | 100 | — | 74 | 18 | 95 |
| click-tab-2-easy | 99 | 77 | 99 | 99 | 61 | — | — | — | — | — | — |
| click-tab-2-hard | 97 | 0 | 96 | 98 | 19 | — | — | — | 76 | 12 | 95 |
| click-tab-2-medium | 100 | 7 | 97 | 99 | 54 | — | — | — | — | — | — |
| click-test | 100 | 100 | 100 | 100 | 100 | 100 | 100 | — | 100 | 100 | 100 |
| click-test-2 | 100 | 100 | 99 | 100 | 95 | 100 | 100 | — | 100 | 100 | 100 |
| click-test-transfer | 100 | 100 | 99 | 100 | 94 | — | — | — | — | — | — |
| click-widget | 100 | 87 | 83 | 100 | 56 | 93 | 100 | — | 98 | 100 | 100 |
| count-shape | 70 | 0 | 82 | 85 | 21 | 76 | — | — | 40 | 41 | 68 |
| count-sides | 100 | 38 | 98 | 100 | 74 | — | — | — | — | — | — |
| drag-box | 99 | 100 | 99 | 100 | 61 | — | — | — | — | — | — |
| drag-item | 100 | 85 | 98 | 100 | 61 | — | — | — | — | — | — |
| drag-items | 100 | 64 | 93 | 99 | 13 | — | — | — | — | — | — |
| drag-items-grid | 89 | 60 | 87 | 98 | 5 | — | — | — | — | — | — |
| drag-shapes | 98 | 96 | 96 | 99 | 26 | — | — | — | — | — | — |
| drag-sort-numbers | 95 | 8 | 92 | 97 | 11 | — | — | — | — | — | — |
| email-inbox-delete | 100 | 99 | 99 | 100 | 22 | — | 100 | — | — | — | — |
| email-inbox-important | 100 | 99 | 99 | 100 | 30 | — | — | — | — | — | — |
| enter-date | 100 | 59 | 97 | 100 | 2 | 96 | — | 100 | 96 | 0 | 100 |
| enter-text-2 | 97 | 100 | 91 | 98 | 4 | — | — | — | — | — | — |
| enter-time | 100 | 78 | 98 | 97 | 4 | 90 | — | — | 100 | 0 | 0 |
| find-midpoint | 96 | 74 | 94 | 97 | 35 | — | — | — | — | — | — |
| grid-coordinate | 92 | 97 | 87 | 100 | 66 | 100 | — | — | 100 | 49 | 100 |
| identify-shape | 100 | 94 | 98 | 100 | 68 | 100 | — | — | 76 | 88 | 100 |
| navigate-tree | 99 | 7 | 98 | 99 | 32 | 99 | 100 | 100 | 86 | 91 | 100 |
| number-checkboxes | 84 | 26 | 96 | 99 | 0 | — | — | — | — | — | — |
| resize-textarea | 99 | 100 | 94 | 100 | 27 | — | — | — | — | — | — |
| right-angle | 97 | 100 | 87 | 98 | 26 | — | — | — | — | — | — |
| simple-algebra | 100 | 99 | 86 | 75 | 3 | — | — | — | 100 | — | — |
| simple-arithmetic | 100 | 67 | 96 | 86 | 38 | — | — | — | — | — | — |
| text-transform | 92 | 91 | 86 | 60 | 19 | — | — | — | 80 | — | — |
| tic-tac-toe | 83 | 76 | 71 | 83 | 32 | 47 | — | — | 56 | 48 | 56 |
| unicode-test | 100 | 64 | 99 | 100 | 86 | — | — | — | — | — | — |
| use-autocomplete | 99 | 95 | 98 | 100 | 7 | 98 | — | — | 58 | 22 | 98 |
| use-colorwheel | 97 | 98 | 90 | 98 | 68 | — | — | — | — | — | — |
| use-colorwheel-2 | 95 | 100 | 94 | 95 | 38 | — | — | — | — | — | — |
| use-slider | 92 | 69 | 98 | 91 | 18 | — | — | — | — | — | — |
| use-slider-2 | 100 | 9 | 97 | 95 | 3 | — | — | — | — | — | — |
| visual-addition | 100 | 68 | 97 | 99 | 36 | — | — | — | — | — | — |
| average | 96.2 | 66.5 | 94.3 | 96.3 | 38.7 | — | — | — | — | — | — |

Table 4: Mean scores across 59 MiniWob++ tasks.

