# OpenReview forum: "From Pixels to UI Actions: Learning to Follow Instructions via Graphical User Interfaces"
_NeurIPS.cc/2023/Conference — NeurIPS 2023 spotlight_

### Official Review · Reviewer_uTRS · 2023-06-30

**Soundness:** 3 good
**Presentation:** 2 fair
**Contribution:** 3 good
**Rating:** 6
**Confidence:** 3

**Summary:**

The paper describes a system to perform pixel-based interaction with human-taylored screens based on GUIs. The proposed system relies on a transformation to a text-based, structured representation of the screen and the screen tasks performed using a larger, pre-trained model of almost 300M parameters. The proposed system achieves performance similar to human beings performing those tasks and SOTA using text representations of the screen.

**Strengths:**

The main strengths of the paper are:

1. It demonstrates a successful system able to interact with the screen (web-based applications)  in a similar way that humans do.
2. It does so by fine-tuning a pre-trained system with a small number of demonstrations.
3. It performs an ablation test where it shows that the pre-training information is critical.

**Weaknesses:**

The main weaknesses of the paper are:

1. The evaluation procedure is so succinctly described that it is hard to understand what was exactly done.
2. The granularity of number of tests according to test type seems to be too coarse (500 tests for 59 task is less than 10 per task).
3. Results are report with little extra information such as mean scores.
4. No statistical significance analysis is performed justifying the claims that the scores are better than SOTA or human (therefore here I assume they are similar, not better).
5. The discussion section is very limited and not clear about the actual contributions.
6. The authors fail to discuss important ethical issues concerning their research. In particular, if AI systems can interact using GUI interfaces, they can produce content and affect web-based systems in very impactful ways, including altering political and social discourses, create hard-to-identify denial attacks, fool "I am not a robot" tests, and pose as people while interacting with real people.

**Questions:**

1. Are the results of the proposed system statistically better than the other systems? With what level of confidence?
2. Can you provide the standard deviations of the tests performed? Are they adequate?

**Limitations:**

The limitation section is very brief and not adequate for a paper with the ethical impacts that such technologies can create.

---

> ### Author Rebuttal · Authors · 2023-08-09
>
> Thank you for your review!
>
> > Ethical considerations
>
> We agree there are important considerations for responsibly developing and deploying models that can interact with websites. While we attempted to identify some of these concerns (e.g. breaking CAPTCHAs) in the “Broader Impact” subsection of Section 7, you have raised several important additional concerns. We will expand the discussion of these issues, and potential mitigations, in our revised paper. See also our response to Ethics Reviewer Vxcz and Ethics Reviewer n367.
>
> > “The evaluation procedure is so succinctly described that it is hard to understand what was exactly done. The granularity of number of tests according to test type seems to be too coarse (500 tests for 59 task is less than 10 per task).“
>
> Please note that there are two paragraphs that discuss evaluation separately for MiniWob++ (Section 4.1) and WebShop (Section 4.2). For MiniWob++, we evaluate on 100 random instances *per task*, for a total of 5900 instances. For WebShop, we follow the standard procedure of evaluating on the 500 test instances. In both cases, our evaluations are consistent and comparable with prior work. We will try to provide additional context for readers unfamiliar with MiniWob++ and WebShop, for our revised paper.
>
> > “Results are report with little extra information such as mean scores.”
>
> We report mean scores across test instances as the primary metric, as described in the “Evaluation” paragraphs of sections 5.1 and 5.2. We also provide more detailed per-task scores in Table 5 of Appendix C.2.
>
> > Statistical significance analysis
>
> For MiniWob++, we estimated variance by computing the mean score for 3 different sets of random seeds for generating test instances. This yielded mean scores of 96.2, 96.4, and 96.1; the standard deviation across these trials was 0.15. These results will be added to Appendix C for our revised paper.
>
> The key comparison for Pix2Act on our proposed setting for MiniWob++ is with CCNet without DOM information, where Pix2Act outperforms with a mean score of 96.2 compared to 24.1.

---

> > ### Comment · Reviewer_uTRS · 2023-08-14
> >
> > Thanks to the authors.
> >
> > I confirm I have read the rebuttal.

---

### Official Review · Reviewer_X3Sy · 2023-07-03

**Soundness:** 3 good
**Presentation:** 2 fair
**Contribution:** 2 fair
**Rating:** 4
**Confidence:** 4

**Summary:**

This paper investigates to create agents that interact with the UIs based on pixel-based screenshots and a generic action space corresponding to keyboard and mouse actions. Extensive experiments on simulated environments, i.e., MiniWob++, WebShop, evaluate the effectiveness of proposed agents, show the benefits of pretraining, and demonstrate the successful application of tree search.

**Strengths:**

This paper builds a pixel-based agent for instruction-following interaction with UIs. This is an interesting and valuable research topic of high bussiness impacts as well for building AI-based UI interaction assistants. The proposed methods are reasonable, and the conducted experiments are clearly presented. The corresponding analysis and discussion for limitations are comprehensive and sufficient.

**Weaknesses:**

The biggest problem of this work is about the novelty of its proposed method. It's kind of difficult for readers to catch up with the core differences compared with the existing pixel-based agents, e.g., Pix2Act, and understraning its advantages. The authors should provide mode insight by highlighting the novelty of proposed methods and clarifying the rationale behind its advantages.

As for the application of tree search, it is widely seen in RL-based agent building. The corresponding introduction in this works, including its motivation, formulation and benefits, lacks of a detailed and clear statement.

**Questions:**

1. What are the key differences between this work and the existing pixel-only UI agents and tree search based RL agents in terms of the methodology?
2. What are the motivations, detailed formulation and beneits of applying tree search here? Are there anything special for completing UI tasks?

**Limitations:**

Pls kindly see the contents of weakness and questions as above.

---

> ### Author Rebuttal · Authors · 2023-08-09
>
> Thank you for your review!
>
> > “The biggest problem of this work is about the novelty of its proposed method. It's kind of difficult for readers to catch up with the core differences compared with the existing pixel-based agents, e.g., Pix2Act, and understraning its advantages.”
>
> We are confused by this comment. This paper introduces Pix2Act. Pix2Act is not an approach from prior work.
>
> The only comparable pixel-based agent from prior work for the tasks that we study was a version of CCNet proposed by Humphreys et al. 2022. We discuss CCNet in the introduction, and offer several empirical comparisons in Section 5. One of the key differences is that Pix2Act is pre-trained on web-scale data, which we find demonstrates significant transfer to the instruction following tasks that we study. Pix2Act significantly outperforms the pixel-only version of CCNet on MiniWob++ with a mean score of 96.2 vs. 24.1 (Figure 3).
>
> Perhaps you are interested in the differences between Pix2Act and Pix2Struct? The underlying image-to-text model for Pix2Act is initialized from the pre-trained Pix2Struct model of Lee et al. 2023, but Pix2Struct image-to-text models have not previously been used to develop pixel-based agents. We will attempt to clarify this relationship in the introduction of Section 3.
>
> > “As for the application of tree search, it is widely seen in RL-based agent building.” “What are the motivations, detailed formulation and beneits of applying tree search here? Are there anything special for completing UI tasks?”
>
> We agree that tree search has been used by prior work in the context of games and other environments, and we did not intend to claim any conceptual novelty in the MCTS algorithm that we apply. Indeed, we adopted MCTS because it is a well-studied algorithm and “has been successfully integrated with neural network policies in prior work” (Section 3.1). We will provide some additional context of prior work related to tree search for policy improvement to our revised paper. The formulation of our MCTS implementation, including integration with policy and value networks, is detailed in Appendix B. We hope the success of tree search on the tasks that we study will inspire future work to consider related methods.

---

> > ### Comment · Reviewer_X3Sy · 2023-08-18
> > **Thanks for your response**
> >
> > Thanks for your response and sorry for the confusion. I made a typo there. The related work I would like to list is Seq2Act [1] rather than Pix2Act. Besides, there is another work for developing pixel-based agents, named Spotlight [2]. Please clarify the relations and differences compared to them.
> >
> > [1] Li, Yang, et al. "Mapping natural language instructions to mobile UI action sequences." arXiv preprint arXiv:2005.03776 (2020).
> > [2] Li, Gang, and Yang Li. "Spotlight: Mobile UI understanding using vision-language models with a focus." (2023).

---

> > > ### Author Response · Authors · 2023-08-18
> > >
> > > Thank you for clarifying your comment.
> > >
> > > The key difference between Pix2Act and the Seq2Act model of Li et al. (2020) is that Seq2Act is *not pixel-based*. Seq2Act represents screens using text-based information corresponding to the Android view hierarchy (which is similar to DOM information for web pages). Li et al. (2020) only briefly discuss the possibility of using pixel-based inputs in the future, mentioning that “while it is possible to directly use screen visual data for grounding, detecting UI objects from raw pixels is nontrivial”. In contrast, in our paper, we focus on agents which do not have access to or rely on structured representations such as the Android View Hierarchy, but instead rely on pixel-based representations of the input screen.
> > >
> > > The Spotlight paper of Li et al. (2023) *does not present an agent that interacts with an environment*. Their paper focuses on various supervised learning benchmarks related to user interfaces, similarly to Pix2Struct (albeit with a narrower focus in terms of tasks). Spotlight performs comparably to Pix2Struct (141.8 vs 136.7 on Widget Captioning, and 106.7 vs 109.4 on Screen2Words) on the tasks on which both models were evaluated. Therefore, Spotlight could potentially serve as an alternative underlying pre-trained model for an agent such as Pix2Act instead of Pix2Struct.
> > >
> > > We hope this clarifies the relationship between Pix2Act and these other prior works. We briefly mention the Seq2Act paper in the related work section of our submission, but we will expand our discussion of both of these two papers in our revised paper.
> > >
> > > Please let us know if there is anything else we can provide or clarify. We are happy to see the strengths mentioned in your original review, and hope we have addressed some of the weaknesses.

---

> > > > ### Comment · Reviewer_X3Sy · 2023-08-21
> > > > **Further Question**
> > > >
> > > > Thanks for your clarification. However, regarding the prior work Spotlight, it has provided experiments for showing its ability on command grounding. Why do you think it does not present an agent that interacts with an environment？

---

> > > > > ### Author Response · Authors · 2023-08-21
> > > > >
> > > > > While the Command Grounding task used to evaluate Spotlight can appear similar to the tasks that we study (MiniWob++ and WebShop), the tasks vary considerably in how they are framed and the capabilities required to perform them. Notably, the Command Grounding task *does not require interacting with an environment*, such as a web browser or Android environment.
> > > > >
> > > > > We use the terms “agent” and “environment” in the sense defined by the Sutton & Barto (1998): “The learner and decision-maker is called the agent. The thing it interacts with, comprising everything outside the agent, is called the environment. These interact continually, the agent selecting actions and the environment responding to those actions and presenting new situations to the agent.” In the Command Grounding task, there is no “environment” in this sense, and there is no “agent” that receives new observations in response to its actions. Indeed, the term “agent” does not appear in the Spotlight paper, and no “environment” is described.
> > > > >
> > > > > Further details on the Command Grounding dataset can be found in section 3.4 of https://arxiv.org/abs/2112.05692, which introduced the dataset. The dataset consists of a fixed set of 1432 unique screenshots from various Android apps. For each screenshot, a set of objects is provided (via the Android View Hierarchy). There are 10,000 natural language instructions that reference one of the objects from the set of objects for a given screen. Given the input image, the set of objects, and the natural language instruction referencing one of these objects, the task requires selecting an object from this set. This fixed dataset is split into a train, dev, and test set.
> > > > >
> > > > > In the Spotlight paper, the task is essentially framed as a binary classification problem. From section 4 of that paper: *”For command grounding and tappability prediction, given a target region, we ask the model to predict either Yes or No tokens following a prompt text, which can be a grounding command or a question for tappability. For command grounding, we let the model inspect each object on the screen by predicting Yes or No following a given command, and the object that yields the largest probability for the Yes token is used as the prediction. During both pretraining and finetuning, the entire Spotlight model is trained end to end by minimizing the cross entropy loss… “*
> > > > >
> > > > > We also note that the Command Grounding task does not assess all of the capabilities required to perform instruction following tasks such as MiniWob++ and WebShop. For example, the Command Grounding task only involves selecting an object on the screen, but does not specify the type of interaction, such as clicking or typing. Also, the instructions used in the Command Grounding task were collected such that they refer to a specific object on the screen. Therefore, there are no instructions that require completing more complex multi-step interactions within an environment.
> > > > >
> > > > > We hope this clarifies some of the key differences between the Command Grounding task used in the Spotlight paper vs. the tasks that we study (MiniWob++ and WebShop).

---

> > > > > > ### Comment · Reviewer_X3Sy · 2023-08-21
> > > > > > **Further Question**
> > > > > >
> > > > > > Thanks much for the discussion. And I would like to suggest you to include these analysis in the revised paper. Could you please highlight/summarize the fundamental difference between them?
> > > > > >
> > > > > > From my side, the command grounding dataset seems more realistic while MiniWob++ and WebShop are the simulation environments over-simplifying the real UI environments. Which task is more challenging and Why?

---

> > > > > > > ### Author Response · Authors · 2023-08-21
> > > > > > >
> > > > > > > > Thanks much for the discussion. And I would like to suggest you to include these analysis in the revised paper. Could you please highlight/summarize the fundamental difference between them?
> > > > > > >
> > > > > > > Thanks, yes, we will summarize our discussion and add it to the related work section of our revised paper!
> > > > > > >
> > > > > > > > From my side, the command grounding dataset seems more realistic while MiniWob++ and WebShop are the simulation environments over-simplifying the real UI environments. Which task is more challenging and Why?
> > > > > > >
> > > > > > > We acknowledge that MiniWob++ and WebShop have limitations with respect to how well they reflect real-world websites and user needs (see also our responses to Reviewer 5HMX and Reviewer CAUm). We think that developing increasingly realistic environments for training and evaluating GUI-based instruction following agents is an important direction for future work. However, we note that MiniWob++ (and, more recently, WebShop) have been widely used by prior work to study instruction following within GUI-based environments. These datasets cover a wide range of instruction types and capabilities, requiring relatively complex sequences of mouse and keyboard actions in order to successfully follow various instructions.
> > > > > > >
> > > > > > > In contrast, as we have described, the Command Grounding task does not involve any environment interaction. By design, the instructions in the dataset always correspond to selecting a single UI element on the given screen. Many of the key aspects of our approach would not be required in this simpler setting, such as learning from multi-step human demonstrations, learning from environment feedback via tree search, executing different action types such as typing and dragging, and evaluating agents via environment interaction and reward signals. The Command Grounding dataset also does not appear to be publicly released. The Spotlight paper shares authors with the paper that originally proposed the dataset (https://arxiv.org/abs/2112.05692), and is the only subsequent work that has adopted this dataset that we have found.
> > > > > > >
> > > > > > > We hope that this discussion, and adding a summary of it to the related work section of our paper, has resolved your concerns about how our work relates to Seq2Act, Spotlight, and the Command Grounding dataset.

---

> > > > > > > > ### Author Response · Authors · 2023-08-21
> > > > > > > >
> > > > > > > > In summary, we want to thank you for your suggestions which we will incorporate into our revised paper, and we hope you would consider increasing your overall rating.

---

### Official Review · Reviewer_sYmC · 2023-07-04

**Soundness:** 3 good
**Presentation:** 3 good
**Contribution:** 3 good
**Rating:** 6
**Confidence:** 4

**Summary:**

One of the main goals of intelligent agents is to interact with the internet in the same way that humans do. Prior state-of-the-art models relied on both the DOM structure and the graphical user interface (GUI) to achieve good performance on web browsing tasks that involve following instructions. In this paper, the authors propose a model that relies solely on pixel-based screenshots as input. The model then selects actions that correspond to basic mouse and keyboard operations.

The authors use a pre-trained model called Pix2Struct and a standard behavior cloning and reinforcement learning (RL) framework to improve the model's ability to follow instructions and perform various web tasks. The authors demonstrate that their pixel-based approach, which does not require access to DOM-related information, can achieve competitive performance on the MiniWob++ benchmark. Additionally, they show through ablation studies that the performance gains can be primarily attributed to the pre-training of Pix2Struct.

**Update after rebuttal**
I confirm that I have read the rebuttal and other reviewers feedback. Authors have addressed my concerns.

**Strengths:**

- Teaching agents to interact with the web is an important research direction. The experiments presented in this paper provide valuable insights to the community. In particular, the authors demonstrate that large-scale pre-training on image-to-text tasks, such as Pix2Struct, can reduce the need to explicitly provide DOM information to the model. Additionally, the task transfer ablation study shows that the model is able to generalize to other tasks. These findings will help to shape the future research direction on this topic.

- Paper is well written and is easy to follow. For MiniWob++ benchmark, authors performed sufficient ablation studies to quantify gains due to individual components i.e. pre-training, behavior cloning.

**Weaknesses:**

- It's not clear if authors have tuned all baselines correctly. In particular, authors show that their proposed RL data bootstrapping method leads to better performance than Behavior Cloning. If I understood the experiments correctly, they have not trained the model for the same number of steps so it's very likely that behavior cloning baseline is potentially under-trained.
- There are multiple important design discussions that are added to the text without any particular justification. For example, authors fine-tune model on MiniWoB++ data for WebShop experiment but don't use WebShop data for MiniWob++ experiment. There is no justification for such design choices. Further, authors provide different learning rate and number of steps for various experiments without justifying these design choices.
- Authors present very sparse results and discussion on WebShop dataset and all experiments are mainly conducted on MiniWoB++ dataset. As a reader, it feels like that authors added that dataset mainly for the sake of adding it.

**Questions:**

- Hyperparameter on Webshop data: learning rate and optimizer details. If these are similar to MiniWoB++, please add a note
- Choose-date: what kind of mistakes model make ? How does RL sampled data help with poor performing tasks?
- Fig 3: for how many steps did you train Ours (BC only) vs Ours ? In particular, does BC training for additional steps help in improving BC baseline?
- Authors suggest that intermediate fine-tuning on MiniWoB++ helps for WebShop tasks. Do they observe similar gains while using WebShop data for MiniWoB++ task?

**Limitations:**

Authors have addressed major limitations of their work and have also provided potential misuse of their research.

---

> ### Author Rebuttal · Authors · 2023-08-09
>
> Thank you for your review!
>
> > WebShop details and hyperparameters
>
> While some prior work (e.g. Humphreys et al. 2022) evaluated only on MiniWob, we believe it was important to also evaluate on WebShop to better understand the generality and limitations of our approach. We will provide some additional details, discussion, and error analysis for WebShop as space allows in our revised paper.
>
> Thank you for pointing out the missing optimizer hyperparameters for WebShop. We indeed use the same optimizer and learning rate as for MiniWob++. We will clarify this in our revised paper in section 5.1.
>
> > Tuning of BC-only model
>
> We did tune the BC-only model to determine the optimal number of training steps. With 26K training steps on MiniWob++ the model completes multiple epochs over the human demonstrations. Training for more steps on the human demonstrations does not improve performance, and can reduce performance due to overfitting.
>
> Notably, our BC-only result is also strong even relative to prior work that uses DOM information (66.5 vs. 38.7 for Humphreys et al. 2022). Finally, the finding that leveraging environment interaction improves performance beyond BC on MiniWob is consistent with prior work that has applied RL methods to MiniWob++ (e.g. Humphreys et al. 2022).
>
> > Using MiniWob data for WebShop and vice versa
>
> The improvement from using MiniWob data for WebShop is 4.0 points in task score (the relevant ablation is mentioned in section 5.4). While perhaps not a key result in the paper, we thought this improvement was significant enough to report.
>
> We did not observe any improvement from using WebShop data for MiniWob. Notably, WebShop has far fewer human demonstrations (1.5K vs. ~15K per MiniWob task), which may explain this asymmetry.
>
> > Error analysis for “choose-date”
>
> This task requires using a calendar interface to select a specific date. The calendar may be initialized to a month (e.g. June) that is far from the target month (e.g. December). The agent must navigate through the calendar one month at a time. A common error is that the model attempts to navigate the calendar in the wrong direction. Perhaps this implies a lack of knowledge related to the ordering of months, or an inability to robustly apply this knowledge in this context.
>
> > How does RL sampled data help with poor performing tasks?
>
> The tree search algorithm can find examples of successful trajectories where the current policy would otherwise fail. Therefore, training on these trajectories can improve performance.

---

> > ### Comment · Reviewer_sYmC · 2023-08-19
> > **Reviewer response**
> >
> > Thanks for answering my questions. Please revise the paper by incorporating additional experiments details.

---

> > > ### Author Response · Authors · 2023-08-21
> > >
> > > Thank you again for your review! We will incorporate additional experimental details in our revised paper.

---

### Official Review · Reviewer_CAUm · 2023-07-21

**Soundness:** 3 good
**Presentation:** 3 good
**Contribution:** 3 good
**Rating:** 7
**Confidence:** 4

**Summary:**

This paper presents PIX2ACT, a method that interacts with GUIs using pixel-level visual representations and generic low-level actions, emulating how humans interact with these interfaces. Unlike previous approaches, it doesn't rely on structured text-based data sources, but rather processes pixel-based screenshots, circumventing issues associated with obfuscation or misalignment in structured application data. The model showed improved performance compared to human crowdworkers on the MiniWob++ benchmark for GUI-based instruction following tasks.


**Strengths:**

The paper introduces a novel approach to the challenge of automated GUI interactions, using pixel-based screenshots as opposed to relying on text-based representations.
The PIX2ACT model was tested on two benchmark datasets, MiniWob++ and WebShop, which ensured a robust and varied testing process.
The paper is clearly written and offers substantial and important contributions, namely the ability to build an agent that can outperform humans in task completion using pixel-based inputs and a generic action space.
Their findings indicate that the pre-training of PIX2STRUCT via screenshot parsing is effective for GUI-based instruction following with pixel-based inputs.


**Weaknesses:**

The performance on the WebShop benchmark is still significantly below larger language models using HTML-based inputs and task-specific actions.
Since the method uses tree search, it seems to rely on offline environments. It’s not clear if this approach will be useful in real-world online environments, although I could see this working with perhaps re-settable virtual environments.


**Questions:**

Would the model maintain high performance in real-time environments, given the training was conducted in offline environments?
How much of the performance difference is due to the size of the model? Can this method be easily adapted to take advantage of existing larger pretrained models?


**Limitations:**

The tree search approach relies on the ability to generate new environment and instruction variations and receive reward signals, which might not be feasible in some real-world applications.
The paper does not provide a clear solution for the performance gap on complex tasks and environments, like the WebShop benchmark.

---

> ### Author Rebuttal · Authors · 2023-08-09
>
> Thank you for your review!
>
> > Limitations of tree search in real-world environments
>
> We tried to address some of the tree search limitations you mentioned in section 7, as well as some potential directions towards applying such an approach more broadly (e.g. generative models of potential instructions and approximate reward models). We will add that our tree search method also requires environments with the ability to reload an initial state, which we agree may not hold in all real-world environments. That said, there are also additional considerations (e.g. security) that may discourage training-time exploration beyond controlled environments in practice.
>
> > Would the model maintain high performance in real-time environments, given the training was conducted in offline environments?
>
> While MiniWob++ and WebShop collectively evaluate a range of capabilities, we hope increasing interest in the community to develop more realistic benchmarks will enable better studying this question in future work.
>
> > How much of the performance difference is due to the size of the model? Can this method be easily adapted to take advantage of existing larger pretrained models?
>
> We have only evaluated Base Pix2Struct models (282M parameters) in this paper. However, conceptually, our method relies only on a pre-trained model implementing a generic image-to-text interface. Therefore, exploring larger pre-trained models is a great direction for future work, especially as such multimodal models become increasingly available.

---

> > ### Comment · Reviewer_CAUm · 2023-08-14
> > **Reviewer Response**
> >
> > Thanks for clarifying these questions. After reading the other reviews I keep my suggestion to accept.

---

### Official Review · Reviewer_5HMX · 2023-07-27

**Soundness:** 3 good
**Presentation:** 3 good
**Contribution:** 3 good
**Rating:** 6
**Confidence:** 4

**Summary:**

The paper introduces PIX2ACT, a model designed to interact with GUIs using only pixel-level visual representations. Unlike most prior works that depend on structured interfaces (like HTML or DOM trees), PIX2ACT relies solely on what it visually sees. This approach is motivated by the way humans interact with interfaces without necessarily knowing the underlying code , which is interesting. The study reveals that such a model can efficiently operate in GUI tasks with generic mouse and keyboard actions, and even outperforms humans in specific benchmarks. For this they use PIX2STRUCT pre-trained on mapping screenshots to structured representation.

Major Contributions:

1. Demonstrated that an agent with pixel-only inputs and generic actions can outperform humans on the MiniWob++ benchmark, achieving performance similar to top-tier agents with access to DOM information.

2. Adapted the WebShop benchmark to function with pixel-based observations and generic actions, establishing the initial baseline performance in this setting.

3. Highlighted the efficiency of PIX2STRUCT’s pre-training through screenshot parsing for GUI-based instruction with pixel-only inputs, leading to notable performance improvements on benchmarks like MiniWob++ and WebShop.

4. Successfully applied tree search for policy improvement in the MiniWob++ environment.

**Strengths:**

The paper studies a new approach to controlling GUIs using just pixel level information unlike alternate approach which uses DOM tree or HTML. While both has its pros and cons , this is one of the new papers ton explore the pixel based controls in this setting.

Adaptable Benchmarking Capabilities: Authors successfully adapt and operate on the WebShop benchmark using solely pixel-based observations and broad actions, laying down a foundational performance standard.

Robust Pre-training Mechanism: The strength of PIX2STRUCT’s screenshot parsing pre-training is evident. This pre-training strategy remarkably boosts the model's efficiency in GUI-based instruction tasks using pixel-based inputs, as observed in dramatic performance jumps on benchmarks. The authors have performed and demonstrated the importance of this through the ablation studies. This is very interesting.

Effective Policy Enhancement with Tree Search: The model employs tree search effectively, which proves to be a straightforward yet potent method for enhancing its policy in environments like MiniWob++.

A very good point that the paper has made - " Finally, aligning human demonstrations with task-dependent actions is often challenging."

**Weaknesses:**

The paper talks about generalization but slightly fails to highlight the same- for example : The chosen datasets ( miniwob++ and webshop(which is a great start) ) are relatively simple datasets and does not capture not capture the full complexity of real-world GUIs.
Addtionally , to perform well reasonable well on the webshop dataset the model seems to require a fine tuning on miniwob++ which is still unclear as to why it is needed. Thus does it actually generalize to webshop? Furthermore, the authors perform hold out set evaluation on miniwob++. However, the tasks are "click-checkboxes-large, click-color, click-tab-2, click-tab-2-hard, count-shape, drag-shapes, use-color-wheel-2, use-slider-2" which could be similar to tasks in the miniwob++ benchmark train set , for instance "click-tab-2-hard" etc. It would be great to test on more diverse and complicated datasets.

Over-reliance on Pre-training: While the pre-training from PIX2STRUCT seems beneficial, depending heavily on pre-trained models can sometimes limit the adaptability and flexibility of the system.

Human Demonstrations: The model seems to be heavily reliant on human demonstrations for training. Gathering these demonstrations can be time-consuming and might not be feasible for all applications. Plus, the quality of these demonstrations can significantly affect the model's performance.

Scalability: The approach described in the paper works well for small, simple web applications. It is not clear how the approach would scale to larger, more complex web applications.

The limitation I see in the environment setup is the use of discrete bins for mouse coordinates and scroll amounts rather than allowing continuous values. A few potential issues with the discrete coordinate binning:
It may make precise clicking and dragging actions more difficult if the bin sizes are too coarse. The agent would need to learn to chain multiple discrete drag actions to move long distances. The optimal binning resolution may vary across different interfaces and tasks. Finding the right granularity could require environment-specific tuning. Discretization can potentially lead to suboptimal policies compared to allowing continuous coordinates.

Similarly for scroll amounts: Scrolling by discrete bins could be inefficient for long pages compared to direct scrolling by pixel amounts. The scroll bin size would again need tuning based on typical page lengths. Overall, the discrete coordinate and scrolling simplification may be necessary to limit the action space size. But it could negatively impact performance on some tasks compared to an environment with continuous values.

No fine-tuning of visual features. The Vision Transformer weights are frozen, so the model may not learn visual representations best suited for this task.

Minor comments: Despite the visual nature of GUIs, prior work has primarily focused on utilizing structured representations of the user interfaces (such as HTML sources, Document Object Model (DOM) trees, and Android view hierarchies) as well as custom, task-specific representations of high-level actions based on these structured representations. ---- > Missing citations.


**Questions:**

Latency and Computation: Transforming screenshots into structured representations can be computationally intensive, which may lead to latency in real-time applications. So what is the typical time taken? Wanted to hear authors thoughts since , the system is based on interpreting pixel-level information from screenshots. This can be computationally intensive?? and may struggle with dynamic content that changes frequently, or with GUI elements that look visually similar but have different functionalities.

Human Demonstrations: The model seems to be heavily reliant on human demonstrations for training. Gathering these demonstrations can be time-consuming and might not be feasible for all applications. Plus, the quality of these demonstrations can significantly affect the model's performance. Does authors have any thoughts one this?

Scalability: How does PIX2ACT scale with different screen resolutions, GUI complexities, or dynamic content on the screens? The performance of models designed for specific benchmarks might not generalize well to real-world applications. Since even the original PIX2STRUCT paper talks about the impact of resolution.

is there a scope for more sophisticated reward function, such as one that provides the agent with feedback about the progress of the task? also is the cumulative reward that accumulates in the miniwob++ benchmark based on the time taken to solve the task in one episode considered? since the agent receives a reward only at a terminal state. Sparse rewards can make learning more challenging, especially if the agent needs to take a long sequence of actions before receiving feedback?

Cursor Representation: The system manually draws a cursor on the screenshot to indicate the mouse pointer position. This might not capture the full nuance of the cursor's state or type (e.g., a hand cursor vs. an arrow cursor), which could provide additional context to the agent.

Greedy Action Selection: The agent follows a greedy policy by selecting the highest scoring action. Does this kind of approach can be short-sighted and might not always result in the optimal long-term strategy??

Beam Search Limitations: While beam search helps in narrowing down the set of most probable sequences, does it always yield the optimal sequence in practice. in the context longer sequences, where the algorithm might not explore sufficiently outside of its "beam" to find a better solution. It can sometimes prefer shorter sequences over more accurate longer ones??

Future work: An additional pre-training task that could potentially improve the model is predicting affordances of UI elements from screenshots. Affordances refer to the possible interactions that an element supports, like if a button can be clicked, a text box can be typed in, a menu can be opened, etc.


**Limitations:**

The authors have mentioned the limitations ( It was thoughtful to anticipate about CAPTCHA) .
However:
Data Privacy: Since they are using screenshots to interpret and interact with GUIs, there might be concerns about data privacy, especially if sensitive information is displayed on the screen. This should be highlighted.
I understand this would be issue while using DOM/HTML as well but highlighted might lead to responsible adoption.

---

> ### Author Rebuttal · Authors · 2023-08-09
>
> Thank you for your review!
>
> > Discrete bins for coordinates and scrolling
>
> We utilized discrete coordinate bins primarily for simplicity. We agree with the limitations you mentioned. While some prior work has also used coordinate bins (e.g. Humphreys et al. 2022), other work has used regression objectives or relative coordinate bins of varying precision (e.g. Baker et al. 2022, https://arxiv.org/abs/2206.11795), which may be useful to adapt for future work. We will add a discussion of this to the limitations section.
>
> > No fine-tuning of visual features
>
> We do in fact tune all parameters of the underlying model, including both the ViT encoder and the text decoder. We will make this clearer in the revised paper.
>
> > Citations in introduction
>
> Thank you, we will add citations or a forward reference to Section 6 where we discuss such approaches in detail.
>
> > Latency and Computation
>
> We did not focus on optimizing the latency of our model, but we measured the latency of processing a single screenshot and emitting an action to be 0.4s, when run on a single Google Cloud TPU v2. While our model is small (282M parameters) compared to some LLMs, further optimizations would likely be required to run in a real time setting at frame rate greater than 2 frames per second. Indeed, as we mention in section 4.1, to simplify our setting “capturing real-time observations for animated elements” is not supported, but could be interesting to consider for future work.
>
> > “GUI elements that look visually similar but have different functionalities”
>
> Conversely, elements with similar functionality may look visually similar but have different source code implementations across applications. We hope our work encourages further investigation of the pros and cons of representing web pages based on their source code vs. visual rendering.
>
> > Human Demonstrations
>
> While we are building off of and comparing with prior work that used human demonstrations, we agree that reducing the quantity of human demonstrations necessary to achieve strong performance in a new environment is a desirable goal for future work.
>
> > Scalability
>
> While the underlying Pix2Struct model supports variable-sized aspect ratios and has demonstrated success across a range of resolutions, we are unfortunately limited by the evaluations available to us. While MiniWob++ and WebShop collectively evaluate a range of capabilities, we hope increasing interest in the community to develop more realistic benchmarks will enable better studying this question in future work. For example, some benchmarks that have been released very recently (after the Neurips submission date) include Mind2Web (https://arxiv.org/abs/2306.06070) and WebArena (https://arxiv.org/abs/2307.13854).
>
> > Reward function and long trajectories
>
> For MiniWob, consistent with prior work, we do not use time-decayed rewards for evaluating and comparing different approaches. Additionally, the agent receives a reward only in the terminal state. We found that this can indeed cause challenges, e.g. for finding a correct trajectory using tree search for tasks that require longer trajectories. Perhaps related to your suggestion, for MCTS we trained a value network to estimate the future reward, based on a surrogate reward that encourages shorter trajectories. This provides information related to “progress on the task”, and Table 3 in Appendix B demonstrates that using this signal is useful. Appendix B also provides the relevant technical details. This approach to MCTS is not conceptually novel, but to the best of our knowledge this is its first application to the types of tasks we study.
>
> > Cursor representation
>
> We do in fact dynamically change the cursor rendering based on the cursor type according to the `cursor` CSS property of the currently hovered element. This can be seen in Figure 2 step 3, where the color selector element has the css property `cursor=crosshair`. That said, we didn’t find much evidence that this actually improves performance.
>
> > Greedy action selection
>
> Greedy action selection is indeed limited, especially when the underlying model is weaker. This can be seen by the difference between the greedy policy vs. tree search policy (Table 2). However, at test time, we assume the agent cannot revise previously chosen actions so report the greedy policy results for consistency with prior work.
>
> > Beam search limitations
>
> Beam search is used in the text decoder to determine the top-k actions for a given step. Most actions have a similar token length (example action strings are shown in Figure 1). Additionally, following T5 and Pix2Struct, we use length normalization (briefly mentioned in Appendix B.1) to attempt to offset beam search’s bias towards shorter sequences, which we will clarify in the main text.
>
> We also had initial experiments exploring beam search as an alternative to MCTS for identifying high-reward sequences of actions, but it did not perform as well as MCTS, especially for tasks requiring longer trajectories.
>
> > Future work and pre-training
>
> We agree with your proposed direction for future work. This type of affordance information should be abundant on the web and provide a potentially useful pre-training signal.
>
> > Data privacy
>
> Thank you for raising this potential concern. We will incorporate this into the revised paper.

---

> > ### Comment · Reviewer_5HMX · 2023-08-21
> > **Acknowledging Authors rebuttal**
> >
> > I have reviewed the authors' rebuttal and acknowledge the points they have addressed. I appreciate their efforts in clarifying my concerns. I eagerly await the revised version to see the implemented changes. Great work thus far.

---

### Official Review · Reviewer_XzpC · 2023-08-01

**Soundness:** 3 good
**Presentation:** 3 good
**Contribution:** 2 fair
**Rating:** 5
**Confidence:** 2

**Summary:**

This work explores the possibility of building an agent that can complete tasks for users solely based on pixel-level visual representations of the GUI state and generic low-level actions, without relying on structured or task-specific representations. The authors demonstrate the effectiveness of their approach on two benchmarks, MiniWob++ and WebShop, adapted to their general Chrome-based environment framework.

**Strengths:**

This submission demonstrates significant strength in several key areas. Firstly, it presents a clear and compelling motivation for developing an agent capable of directly interacting with pixel-level GUI interfaces, aligning with a more natural and human-like manner. This aspect alone distinguishes the submission and makes it particularly appealing to the readers.

Moreover, the use of well-crafted illustrations enhances the understanding of the content and effectively communicates the main ideas, further solidifying its strength. Readers can easily grasp the concepts presented, adding to the submission's overall impact.

Furthermore, the proposed framework is rigorously validated on two datasets, and the comparison with baselines using DOM/HTML as input is conducted in great detail. This thorough analysis strengthens the submission's credibility and highlights its robustness.


**Weaknesses:**

I have several concerns regarding the experimental design in the submission:

(1) One notable issue is the absence of certain ablations that would provide valuable insights into the effectiveness of each component in the proposed agent framework. Specifically, the authors should address the bottleneck of the framework, whether it lies in the visual encoder or the text decoder. Additionally, it would be beneficial to explore the potential performance improvements by utilizing alternative variants of visual encoders or text decoders. Furthermore, since the visual encoder plays a crucial role in extracting and understanding instructions embedded in the interface, it is essential to verify the effectiveness of using ViT in OCR accuracy.

(2) The rationale behind embedding instructions into the UI screenshot instead of providing them directly as input remains unclear. The submission should elaborate on this choice and justify its benefits. Moreover, considering the availability of open-source Visual Language Models (VLMs) like mPLUG-Owl [1], OpenFlamingo [2], and Otter [3], which natively support instruction following and accept multimodal input, it would be valuable to include these VLMs as baselines in the revised version. This addition would offer a more comprehensive evaluation and better contextualize the proposed method's unique contribution in comparison to existing approaches.

[1] https://github.com/X-PLUG/mPLUG-Owl
[2] https://github.com/mlfoundations/open_flamingo
[3] https://github.com/Luodian/Otter

Addressing these concerns would significantly enhance the experimental design and strengthen the submission's overall validity and contribution to the field.


**Questions:**

(1) In Figure 2, step 3, why are there two overlapping ‘+’? Does it mean the mouse is pressed down?

(2) This is not a serious issue, but some Figures are quite blurry (e.g., MiniWob++ examples in Figure1 and top row in Figure 2), is this due to the way the dataset is constructed?


**Limitations:**

The authors have included discussions of the limitations.

---

> ### Author Rebuttal · Authors · 2023-08-09
>
> Thank you for your review!
>
> > Bottleneck is vision encoder or text decoder?
>
> The text decoder is only responsible for decoding short strings corresponding to the closed set of actions shown in Figure 1. Therefore, it seems reasonable to assume that the ViT encoder is the “bottleneck” towards achieving strong performance. As discussed in the introduction, many of the key challenges are essentially representation learning challenges for the encoder, such as understanding the interface layout, recognizing and interpreting visually-situated natural
> language, identifying visual elements, and predicting their functions and methods of interaction. Our ablations highlight the screenshot parsing pre-training task of Lee et al. 2023 as a critical factor towards improving model performance (Figure 3).
>
> > Architectural variations of Vision Transformers
>
> Pix2Act builds off of the Pix2Struct model of Lee et al. 2023, which uses a standard ViT architecture. We believe their strong results on tasks related to UI understanding (e.g. RefExp, Screen2Words) support the choice of this model as a starting point for pushing the boundaries of a pixel-based digital agent.
>
> > “The rationale behind embedding instructions into the UI screenshot instead of providing them directly as input remains unclear.”
>
> While we agree this method for representing instructions can perhaps seem unorthodox at first, prior work (e.g. Lee et al. 2023) has used this method for incorporating instructions in the input of pixel-only models for tasks such as DocVQA. This was hypothesized to enable better transfer from the pre-training task (see section 2.5 of Lee et al. 2023), and validated by strong empirical results. We will add this justification to the paper.
>
> We also note that while this is the most natural way to incorporate instructions when starting from Pix2Struct, providing the instructions as text may work better for other pre-trained models that are tailored to accepting a text instruction as input.
>
> > ​​Baselines for “Visual Language Models (VLMs) like mPLUG-Owl, OpenFlamingo, and Otter”
>
> We are excited by the increasing public availability of larger and more capable VLM models, and we agree evaluating such models on our proposed setting would be interesting for future work (we include some related discussion in section 7). However, we note that the Neurips 2023 submission deadline was May 17, 2023. To the best of our knowledge, the mPLUG-Owl preprint was released on April 27, 2023. OpenFlamingo was announced on March 28, 2023 and a paper is not yet available. The Otter preprint was released on June 8, 2023.
>
> Additionally, compared to Pix2Struct, VLM models such as mPLUG-Owl, OpenFlamingo, or Otter have not demonstrated as strong a capability for learning representations of inputs that contain visually-situated language such as UI screenshots, as measured on tasks such as RefExp, Screen2Words, etc. Indeed, one of the limitations of mPLUG-Owl is “complex OCR” and the ability to correctly interpret web page screenshots (Figure 12 of mPLUG-Owl paper, ​​https://arxiv.org/abs/2304.14178). In contrast to these publicly available VLMs, GPT-4 has demonstrated strong results on tasks involving visually-situated language, outperforming Pix2Struct on ChartQA and AI2D (https://openai.com/research/gpt-4), but as far as we know support for tuning this model on multimodal inputs is not publicly available, and the details of the model have not been published. Regardless, our ablations (Figure 3) suggest that pre-training on the screenshot parsing task of Lee et al. 2023 is critical to achieve strong performance on the tasks that we study, suggesting that this task or a similar one should be included in the pre-training of VLMs that are used to develop digital agents.
>
> > Figures are quite blurry
>
> We will try to improve the rendering quality, but are limited by the resolution of the MiniWob environment, which is 160 by 210 pixels.
>
> > Figure 2 step 3
>
> The color selection element renders a “+” at the currently selected color. When the pointer is positioned over this element, it is also rendered as a “+”, as this element has the css property “cursor=crosshair”. At step 3 of Figure 2, the pointer is positioned over the currently selected color, leading to two similar “+”s that are slightly offset.

---

> > ### Comment · Reviewer_XzpC · 2023-08-10
> >
> > Thanks for providing the rebuttal response.
> >
> > I acknowledge that I have read the response.

---

### Author Rebuttal · Authors · 2023-08-09

We would like to thank all of the reviewers for their comments and suggestions! We tried to address any questions in the individual responses.

We would also like to respond to the ethical reviewers, as we did not see a way to respond individually to ethics reviews.

__Ethics Reviewer Vxcz__

Thank you for your review!

> Human Demonstrations

We have had previous correspondence with the authors of Humphreys et al. (ICML 2022) and Yao et al. (Neurips 2022). Our use of both datasets is consistent with the permission granted by the authors. We can also confirm that neither dataset contains PII. We will work with both groups of authors to confirm further details of the annotation process.

We agree that datasets that incorporate a more diverse set of instructions and example demonstrations, especially those that reflect the needs of diverse users with different abilities, would be a valuable contribution for future work, and will mention this in our revised paper.

> Broader Impacts

We agree there are important considerations for responsibly developing and deploying models that can interact with websites. While we attempted to identify some of these concerns in the “Broader Impact” subsection of Section 7, you have raised several important additional concerns. We will expand the discussion of these issues, and potential mitigations, in our revised paper draft. Taking your comments into consideration, below are some of our thoughts on these issues.

In this paper we have trained and evaluated models only in offline environments. Responsibly deploying models in an environment where they can interact with online services would require additional considerations. Prior to enabling a model to access a new service, it would be important to sufficiently verify and/or constrain the behavior of the model to ensure that it is consistent with the terms-of-service for that service and does not otherwise cause harm.

There would be many potential risks associated with deploying models that could interact with services in violation of their terms-of-service or otherwise engage in various forms of spam, fraud, or abuse. Examples of such behavior could include impersonating human users, generating harmful content or spam, or engaging in denial-of-service attacks. Models that use the same conceptual interface humans use could potentially be more capable of breaking security defenses (e.g. solving CAPTCHAs) or engaging in forms of spam, fraud, or abuse that are more difficult to detect. It is therefore important for research related to security and techniques for detecting spam, fraud, and abuse to take such potential uses into account.

__Ethics Reviewer n367__

Thank you for your review!

We will elaborate on the ethical considerations related to security, and possible mitigations, in our revised paper. Relevant security research would include work on CAPTCHAs. For example, Section 3.4 “Attacks against Behavior-based CAPTCHA” of  https://arxiv.org/abs/2103.01748 discusses CAPTCHA attacks by bots imitating human behavior, and the authors also discuss potential mitigations.

Please also see our response to Ethics Reviewer Vxcz for a broader discussion of other ethical considerations.

---

### Decision · Program_Chairs · 2023-09-21

**Decision:**

Accept (spotlight)

**Comment:**

Overall I believe this paper should be accepted - I have one weak reject, but all others say accept. The authors and the weak reject have had several exchanges and I believe issues have been resolved but the score has not been changed. I recommended to accept two weeks ago and have received no objections. The authors have addressed ethics concerns in my opinion.

I believe one reviewer summarizes the opinion well with the following:

This submission demonstrates significant strength in several key areas. Firstly, it presents a clear and compelling motivation for developing an agent capable of directly interacting with pixel-level GUI interfaces, aligning with a more natural and human-like manner. This aspect alone distinguishes the submission and makes it particularly appealing to the readers.

Moreover, the use of well-crafted illustrations enhances the understanding of the content and effectively communicates the main ideas, further solidifying its strength. Readers can easily grasp the concepts presented, adding to the submission's overall impact.

Furthermore, the proposed framework is rigorously validated on two datasets, and the comparison with baselines using DOM/HTML as input is conducted in great detail. This thorough analysis strengthens the submission's credibility and highlights its robustness.